

# Constructing Chinese taxonomy trees from understanding and generative pretrained language models

Jianyu Guo, Jingnan Chen, Li Ren, Huanlai Zhou, Wenbo Xu and Haitao Jia

University of Electronic Science and Technology of China, ChengDu, Sichuan, China

## ABSTRACT

The construction of hypernym taxonomic trees, a critical task in the field of natural language processing, involves extracting lexical relationships, specifically creating a tree structure that represents hypernym relationships among a given set of words within the same domain. In this work, we present a method for constructing hypernym taxonomy trees in the Chinese language domain, and we named it CHRRM (Chinese Hypernym Relationship Reasoning Model). Our method consists of two main steps: First, we utilize pre-trained models to predict hypernym relationships between pairs of words; second, we regard these relationships as edges to form a maximum spanning tree in the word graph. Our method enhances the effectiveness of constructing hypernym taxonomic trees based on pre-trained models through two key improvements: (1) We optimize the hyperparameter configuration for this task using pre-trained models from the Bert family and provide explanations for the configuration of these hyperparameters. (2) By employing generative large language models such as ChatGPT and ChatGLM to annotate words, we improve the accuracy of hypernym relationship identification and analyze the feasibility of applying generative large language models to the task of constructing taxonomy trees. We trained our model on subtrees of WORDNET and evaluated its performance on non-overlapping subtrees of WORDNET, demonstrating that our enhancements led to a significant relative improvement of 15.67%, achieving an F1 score of 67.9 on the Chinese WORDNET validation dataset compared to the previous score of 58.7. In conclusion, our study reveals the following key findings: (1) The Roberta-wwm-ext-large model consistently delivers outstanding results in constructing taxonomic trees. (2) Generative large language models, while capable of aiding pre-trained models in improving hypernym recognition accuracy, have limitations related to generation quality and computational resources. (3) Generative large language models can serve various NLP tasks either directly or indirectly; it is feasible to improve the downstream NLU task's performance through the generative content.

Corresponding author
Haitao Jia, jhtao@uestc.edu.cn

# INTRODUCTION

Hyponymy-hypernymy relationship recognition is a classic problem in the field of natural language processing. The hyponymy-hypernymy relationship refers to a semantic inclusion relationship where the concept of a hyponym is subsumed by the concept of a hypernym, such as "red" being a hyponym of "color." Hyponymy-hypernymy relationship recognition involves determining whether two given words have a hyponymy-hypernymy relationship. A taxonomy tree can be constructed by establishing such relationships among a set of words. This taxonomy tree is widely applied in downstream NLP tasks such as text classification, information retrieval, and text generation.

Pre-trained language models are models obtained through unsupervised learning on massive text corpora using pre-training techniques. In recent years, a series of pre-trained models, represented by Bert, have achieved state-of-the-art performance after fine-tuning on various NLP tasks.

Before the emergence of Bert, hyponymy-hypernymy relationship recognition mainly relied on pattern-based methods and statistical or machine learning-based approaches. These methods essentially involved extracting regularities from corpora through a combination of manual pattern summarization and statistical or machine learning methods. Bert significantly improved the accuracy of hyponymy-hypernymy relationship recognition. Based on the Transformer architecture, Bert featured a substantial increase in the number of parameters. However, their interpretability decreased significantly due to the complexity of deep neural network models like Bert. We believe that models like Bert, with multiple attention heads and a massive number of parameters, learn the regularities and knowledge in corpora more comprehensively than manually summarized patterns. Pre-trained models encode domain-specific knowledge as parameters, explaining why they achieve outstanding results in domain hyponymy-hypernymy relationship recognition. Bert models are currently crucial tools for hyponymy-hypernymy relationship recognition. However, Bert outputs logits, which are probability scores. Constructing a taxonomy tree from probability scores requires reconciliation. In this work, logits are treated as edge weights, and a maximum spanning tree algorithm is applied to obtain the taxonomy tree.

Building upon previous work, this work investigates constructing a taxonomy tree in the Chinese domain using pre-trained models based on the authoritative English taxonomy tree WORDNET. By studying the differences among various pre-trained models, optimizing hyperparameter configurations, and adding word glosses, the F1 score for constructing a taxonomy tree in the Chinese domain using pre-trained models increased from 58.3 to 67.9. The article analyzes the impact of various parameters on constructing the taxonomy tree and discusses the feasibility of applying generative language models to taxonomy tree construction.

The structure of this article is as follows: In "Related Work", we introduce related work. In "Methods", we present the specific methods employed in this research. "Experiments" details the experimental process, and "Results and Analysis" provides an analysis the experimental results. In "Conclusions", we present the conclusions of this article and discuss the future development of constructing taxonomy tree task.

## RELATED WORK

Generally, taxonomy tree constructing tasks consist of two parts. The first part involves hyponymy-hypernymy relationship recognition, which is the process of determining whether two given words have a hyponymy-hypernymy relationship. The second part is taxonomy tree reconcile, which aims to organize multiple pairs of domain-specific words with hyponymy-hypernymy relationships into a hierarchical tree structure.

Traditional hyponymy-hypernymy relationship recognition methods are based on patterns or distributed models, which infer hyponymy-hypernymy relationships among words from large corpora. Pattern-based methods, with Herast's work being a prominent example, represent this category. In recent years, *Roller, Kiela & Nickel (2018)* demonstrated that pattern-based models outperformed distributed models in most cases. *Yu et al. (2020)* pointed out that pattern-based extraction methods suffer from sparsity issues. Combining pattern-based and distributed methods within a complementary framework can effectively address sparsity issues and improve performance. *Shang et al. (2020)* proposed a taxonomy construction method for unseen domains. *Pinto, Shraddha Gole & Madasamy (2023)* proposed a novel hypernym discovery algorithm based on pattern. The work noted above still focus on patterns. *Soyalp et al. (2021)* made a summary and pointed models before the Transformer architecture primarily relied on rules, patterns and machine learning. Due to their limited parameter count, these models couldn't fully harness the knowledge in corpora, resulting in inferior extraction performance compared to Transformer models.

*Vaswani et al. (2017)* introduced the Transformer architecture, which remains a fundamental building block for pre-trained models. *Devlin et al. (2019)* introduced the Bert model, which achieved remarkable results on numerous natural language understanding tasks. *Han et al. (2020)* intorduced the impact of context and gloss in relation extraction task. Then, there is a lot of work based on Transformer to solve the problem in hyponymy-hypernymy recognition task. *Chen, Lin & Klein (2021)* proposed the CTP model, which uses understanding pre-trained models to construct taxonomy trees for multiple languages. However, CTP did not fine-tune the parameters extensively, and due to its out-of-vocabulary issues, leaving room for future improvement. *Bai et al. (2022)* introduced HCP, which improved hyponymy-hypernymy recognition performance in context-rich scenes by integrating class-based language models with curriculum learning. *Wang, He & Zhou (2019)* introduced a method based on adversarial learning to improve hypernymy prediction. Besides, *Zhang et al. (2018)*, *Yu et al. (2015)* and *Washio & Kato (2018)* thought word embedding and vector can be useful in taxonomy tree constructing, *Shen & Han (2022)* and *Shen et al. (2018)* proposed a hierarchical tree expansion method based on Data Mining and Knowledge Discovery. These works are all based on large-scale *corpus* and high-dimensional vectors, which is important for large models.

In 2023, generative pre-trained language models gained significant attention. *Brown et al. (2020)* introduced the GPT-3 model, and *Zeng et al. (2022)* presented ChatGLM. *Gilardi, Alizadeh & Kubli (2023)* demonstrates the generative pre-trained language models have the potential to assist humans in data annotation tasks. Some end-to-end, untuned

generative language models have shown competitive performance in some tasks compared to supervised fine-tuned understanding-based models. Past work on understanding and generative models has substantially improved hyponymy-hypernymy relationship recognition's performance, making pre-trained models the mainstream choice for hyponymy-hypernymy relationship recognition.

In the taxonomy tree reconciliation part, *Kozareva & Hovy (2010)* proposed a graph-based longest path method for reconcile taxonomies based on pattern-based hyponymy-hypernymy inference, providing an approach to organize taxonomy trees. *Bansal et al. (2014)* viewed the reconciliation problem as a structured learning problem and used belief propagation to incorporate siblinghood information. *Mao et al. (2018)* introduced a reinforcement learning approach that combined inference and restructuring into a two-stage process.

Pre-trained models for hyponymy-hypernymy relationship recognition output a fully connected weighted graph, decoupling the recognition process from the inference process. The fully connected weighted graph can be used as input for recognition to apply algorithms related to maximum spanning trees. Our work aims to employ an algorithm with a computational complexity lower than $O(n^2)$ for scenarios involving medium-sized word groups. We finally chose the Chu-Liu-Edmonds algorithm to find the maximum spanning tree within word groups.

## METHODS

We named our approach CHRRM (Chinese Hypernym Relationship Reasoning Model), as illustrated in the Fig. 1. The model's input is a set of words, and the output is a taxonomy tree. The orange module constitutes the core component of the method proposed in this article.

### Hypernymy relationship recognition module

For a set $V$ containing $n$ words from some domain, we combine any two words $w_1$ and $w_2$ within the set to construct all relationships between words. This process results in $n * (n - 1)$ relationships. For each of these relationships, we create a sample in the format $[CLS] v_i$ *is a* $v_j [SEP]$ to serve as input to the Bert family models. The output logits are used as weights for each relationship, allowing us to form a fully connected weighted graph.

Due to the constraints of tree structures, in a tree with $n$ nodes, there are $n - 1$ edges. Therefore, for this weighted graph, there should be $n - 1$ positive relationships and $(n - 1)^2$ negative relationships. The transformation of the graph into a tree structure is performed by the Tree Reconciliation Module.

### Tree reconciliation module

The input to the Tree Reconciliation Module is a fully connected directed graph, where the edge weights are calculated by the Hypernymy Relationship Recognition Module. Greater edge weights indicate a stronger belief by the model in the existence of a hypernymy relationship between the two nodes. The output of this module is a maximum spanning

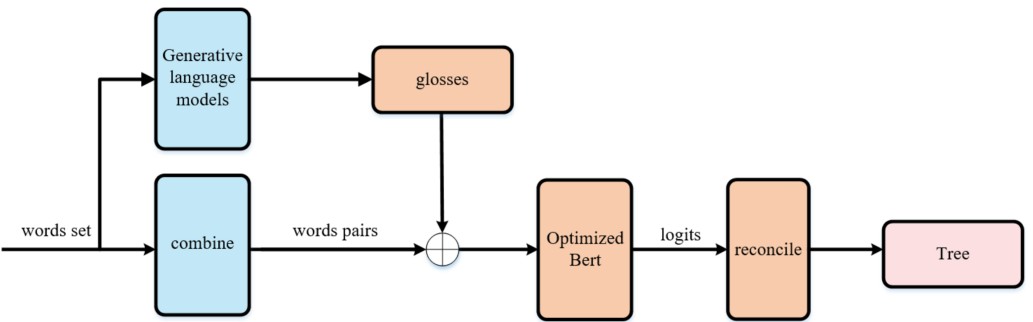

**Figure 1 CHRRM's workflow.** CHRRM's workflow, the model's input is a set of words, and the output is a tree. The orange module constitutes the core component of the method proposed in this article.

tree, which is the taxonomy tree we need. This taxonomy tree is characterized by being connected and acyclic in graph theory terms.

Therefore, we seek a maximum spanning tree algorithm to implement the functionality of this module. Common maximum spanning tree algorithms include the Kruskal algorithm, the Prim algorithm, and the Chu-Liu-Edmonds algorithm (*Chu, 1965*). Among these, the Kruskal and Prim algorithms are suitable for finding maximum spanning trees in undirected graphs, while the Chu-Liu-Edmonds algorithm is suitable for finding maximum spanning trees in directed graphs. The Chu-Liu-Edmonds algorithm greedily selects the maximum incoming edge for each node and contracts any cycles in the tree into a single node, finding the maximum spanning tree in the graph.

After comparing these algorithms, we decided to use the Chu-Liu-Edmonds algorithm to implement the Tree Reconciliation Module because hypernymy relationships are directional and irreversible.

## Generative gloss module

This module investigates the impact of gloss on hypernymy relationship recognition (*Blevins & Zettlemoyer, 2020*). When glosses information is included, the module constructs a prompt for each word w:

*You are a linguistic expert, generate an informative and accessible explanation for the word w.*

This prompt is input to ChatGPT or ChatGLM, yielding the gloss annotation $gloss_w$ for w. Then in hypernymy relationship recognition module, the input is modified to $[CLS]v_i : gloss_{vi}[SEP]v_j : gloss_{vj}[SEP]$ and the logits generated by Bert are used as weights for each relationship.

Subsequently, we employ a controlled variable approach to investigate the performance of different Bert models and generative language models' hyperparameters on the same dataset. The goal is to identify the most suitable parameters for the model within the scope of our research.

Finally, one of trees infered by CHRRM is illustrated in the Fig. 2. It shows taxonomy tree about water bodies.

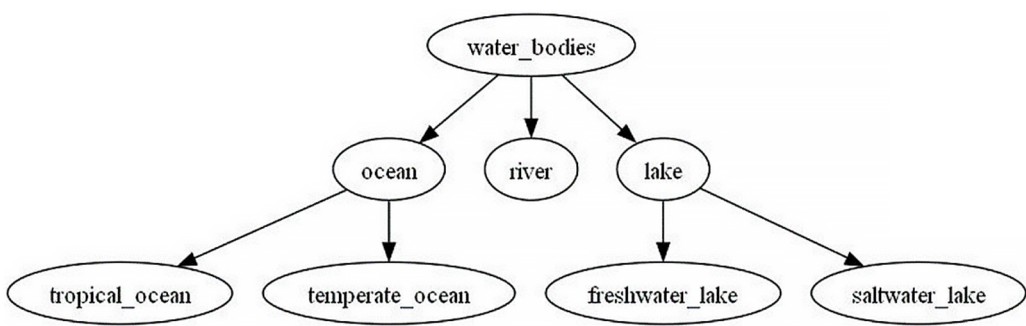

**Figure 2 Water bodies taxonomy tree infered by CHRRM.**

## EXPERIMENTS

In this chapter, we provide an overview of our experimental setup, which includes the dataset, evaluation metrics, runtime parameters, and hardware configuration. All experiments were conducted on one A100 GPU.

### Dataset

The dataset for this task originates from the medium-sized WORDNET subtree dataset introduced by *Bansal et al. (2014)*. The original dataset is in English, and to adapt it for research in the Chinese domain, we performed language alignment. This alignment involved establishing mappings from English to Chinese, guided by the word translations provided by *Wang & Bond (2013)*. However, not all parts of the dataset could be mapped. As a result, we removed tree nodes and their descendants that could not be mapped to ensure that the mapped graph maintained a tree structure. Following these adjustments, we obtained 328 trees, which we split into training, validation, and test sets, each containing 216, 48, and 64 trees with a root node height not exceeding three and the longest path in each tree not exceeding f.our An example of such a tree is illustrated in the Fig. 3, representing a taxonomy tree for the elephant category.

### Evaluation metrics

Following the evaluation standards used in *Bansal et al. (2014)*, we primarily employ the F1 score as our key evaluation metric. The task of constructing taxonomy trees is imbalanced, with a majority of relationships being negative samples. Therefore, the F1 score for positive samples is used as the evaluation metric. In this experiment, we utilize the Ancestor F1, computed as shown in Eq. (3).

$$Precision = \frac{TP}{TP + FP} \tag{1}$$

$$Recall = \frac{TP}{TP + FN} \tag{2}$$

$$F1 = 2 * \frac{Precision * Recall}{Precision + Recall} \tag{3}$$

where TP stands for true positive, denoting cases where the prediction is positive and

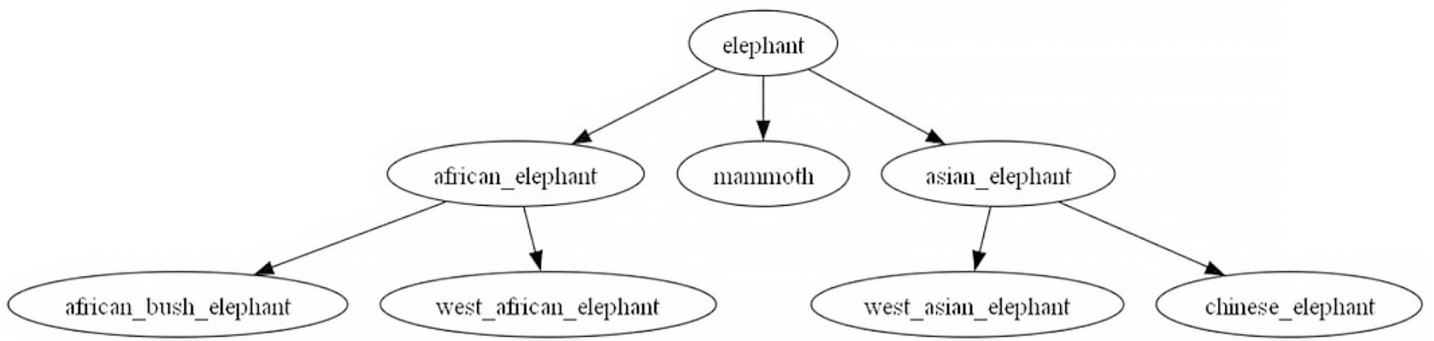

**Figure 3 Elephant taxonomy tree in wordnet.**

**Table 1 Model's primary hyperparameters.**

| Model | Layer | Dimension | Attention head | Parameters |
|---|---|---|---|---|
| Bert-base-chinese | 12 | 768 | 12 | 110 M |
| Hfl-chinese-lert-large | 24 | 1,024 | 16 | 325 M |
| Hfl/chinese-xlnet-mid | 24 | 768 | 12 | 209 M |
| Hfl-chinese-roberta-wwm-ext-large | 24 | 1,024 | 16 | 325 M |

correct. FP represents false positive, indicating instances where the prediction is positive but incorrect. TN refers to true negative, where the prediction is negative and correct, and FN is for false negative, signifying cases where the prediction is negative but incorrect.

## Pre-trained models and hyperparameter design

Our experiments primarily investigate the impact of various Chinese pre-trained models and hyperparameters on the task of constructing taxonomy trees. We selected four representative pre-trained models in the Chinese domain, with the bert-base-chinese model serving as the baseline, the candidates include roberta-wwm-ext-large, lert-large, and chinese-xlnet-mid models. Their primary hyperparameters are listed in Table 1.

**Roberta-wwm-ext-large:** This is an improved version based on Chinese Bert, featuring various enhancements, including the use of dynamic masking strategies, a larger quantity of higher-quality training data, and refined parameter tuning. This model is widely adopted in the Chinese domain and consistently performs exceptionally well across various tasks.

**Lert-large:** This model integrates linguistic features into the pre-trained model based on Chinese Bert. It utilizes three linguistic tasks during pre-training and employs a linguistic-inspired pre-training mechanism. We believe that the task of constructing taxonomy trees benefits from linguistic insights compared to more conventional NLP tasks.

**Chinese-xlnet-mid:** This model is trained on Chinese text data and is based on the XLNet approach. Both XLNet and Bert are models built on the Transformer architecture, but XLNet differs in that it uses a universal auto-regressive pre-training method to address

the performance loss due to inconsistent tasks during fine-tuning, which is associated with the use of masks during Bert pre-training. It features significant improvements compared to models in the Bert family. Therefore, we aim to study its performance in constructing taxonomy trees.

We use the AdamW optimizer and select Learning Rate, Warmup, and Epsilons as the primary changeable hyperparameters for our experiments. The specific values of these hyperparameters are listed in Table 2. According to the study by *Liu et al. (2019)*, we have also chosen some constant hyperparameters that remain unchanged during training, with their specific values detailed in Table 3.

The purpose of the experiment is to study the effect of various parameters on model performance. *Chen, Lin & Klein (2021)* examined the model performance with learning rates of {1e-5, 1e-6, 1e-7}. Since the warmup strategy dynamically affects the learning rate, we included the warmup strategy in our experimental scope based on their work. *Liu et al. (2019)* pointed out that Epsilons significantly affect the training process of the Adam optimizer, therefore, we included Epsilons as one of the changeable hyperparameters. Below is an introduction to each changeable hyperparameter.

**Learning rate:** Learning rate has a significant impact on model's performance. We incrementally increase the learning rate by a factor of two and seek to identify the optimal learning rate.

**Warmup:** Warmup is a common optimization strategy, and in the linear warmup strategy, the learning rate is linearly increased to a peak and then gradually reduced at a lower slope. *He et al. (2016)* has proven this approach can enhance performance.

**Epsilon (eps):** *Liu et al. (2019)* has proven the experiment's sensitivity to the epsilon parameter. Epsilon is primarily designed for numerical stability, preventing anomalies such as divide-by-zero errors during computations.

In summary, combining the experimental designs from related studies, we selected Learning Rate, Warmup, and Epsilons as the three changeable hyperparameters to investigate their effects on Hypernym relationship recognition task.

## Generative annotation design

*Chen, Lin & Klein (2021)* conducted experiments using oracle gloss from WordNet and web gloss from the web to demonstrate the significant impact of gloss quality on the hyponymy-hypernymy relationship recognition task. Generative language models, such as ChatGPT, have the capacity for end-to-end gloss generation. Therefore, we aim to augment the semantic content of vocabulary by employing such models to improve the accuracy of hyponymy-hypernymy relationship recognition.

For two reasons, we did not directly use large-scale generative language models for taxonomy tree construction task. Firstly, generative language models have significantly higher computational requirements than Bert. If every relationship needs to be assessed through question-answering, the computational complexity is $O(n^2)$, leading to substantial time and computational resource consumption. Secondly, generative language models rely on next-token prediction and are highly prompt-dependent, making further research on their capabilities in NLU tasks necessary to maximize their potential and clarify their

**Table 2 Changeable hyperparameters range.**

| Learning rate | Warmup | Epsilon |
| --- | --- | --- |
| 5e-6 | 0.05 | 1e-5 |
| 1e-5 | 0.1 | 1e-6 |
| 1.5e-5 | 0.15 | 1e-7 |
| 2e-5 | 0.2 | 1e-8 |

**Table 3 Constant hyperparameters.**

| Hyperparameters | Value |
| --- | --- |
| Dropout | 0.1 |
| Attention dropout | 0.1 |
| Batchsize | 32 |
| Weight decay | 0.01 |
| Learning rate decay | Linear |
| Adam $\beta 1$ | 0.9 |
| Adam $\beta 2$ | 0.98 |
| Max epochs | 15 |

application. Therefore, we chose to use generative language models for gloss generation as a transitional step to gradually enhance the effectiveness of our model in the taxonomy tree construction task.

## RESULTS AND ANALYSIS

Following the experimental procedures outlined in "Experiments", we achieved significant improvements over previous work. Table 4 is a comparison of the CHRRM model with previous methods, using precision, recall, F1 score metrics on the Chinese WORDNET validation dataset. It is evident that without external gloss, we achieved a 4.0 increase in F1 score compared to previous work, and when external gloss is included, we achieved an additional 5.2 increase. Overall, we achieved a 9.2 improvement, with a relative increase of 16.5%.

After extensive experimentation, we identified a set of optimal parameters. These hyperparameters, when applied to the hfl-chinese-roberta-wwm-ext-large model, yielded an F1 score of 67.9, as shown in the Table 5.

### Pre-trained model selection analysis

In this section, we will compare and evaluate the performance of these Pre-trained models. Table 6 shows the maximum F1 scores obtained by each model under the best parameters with the addition of gloss.

| Table 4 Comparison of the models. | | | |
|---|---|---|---|
| Model | P | R | F1 |
| *Bansal et al. (2014)* | 48.0 | 55.2 | 51.4 |
| *Mao et al. (2018)* | 52.9 | 58.6 | 55.6 |
| CTP | 62.2 | 57.3 | 58.7 |
| CHRRM (no glosses) | 64.5 | 61.0 | 62.7 |
| CHRRM (with glosses) | **73.5** | **63.1** | **67.9** |

| Table 5 Best performance hyperparameters on Roberta. | | | | |
|---|---|---|---|---|
| Model | Learning rate | Epsilon | Warmup | Gloss |
| Hfl-chinese-roberta-wwm-ext-large | 1e-5 | 1e-8 | 0.05 | ChatGPT |

| Table 6 Best performance of pre-trained models. | | | |
|---|---|---|---|
| Model | P | R | F1 |
| Bert-base-chinese | 62.5 | 59.6 | 61.0 |
| Hfl-chinese-lert-large | 68.5 | 65.2 | 66.8 |
| Hfl/chinese-xlnet-mid | 69.9 | 65.1 | 67.4 |
| Hfl-chinese-roberta-wwm-ext-large | 73.5 | 63.1 | 67.9 |

### Bert-base model

The bert-base model, as one of the earliest models released, serves as the baseline in our experiments. Its performance is notably lower than other pre-trained models. We attribute this to two main reasons: (1) Bert's pre-training tasks involve mask prediction and next sentence prediction, which ensures a good understanding of semantics. However, these tasks significantly differ from our downstream tasks, suggesting that the pre-training tasks were a necessary trade-off for Bert's generality. (2) *Liu et al. (2019)* think the Bert model is significantly undertrained, suffering from issues such as improper parameter selection, inadequate training, and low-quality training data. Additionally, the base model itself has only one-third of the parameters of the large model, which limits its ability to accurately model semantics.

### Lert-large model

The lert model, introduced by hfl in 2022, is distinguished by its incorporation of linguistic features. We selected Lert-large because we view the construction of taxonomy trees as a linguistic task that requires logical reasoning and sensitivity to language vocabulary. Lert is considered to have an advantage in this task. Experiments show that Lert, when combined with gloss, uniformly improves precision and recall, ultimately resulting in an F1 score close to the highest achieved.

### Xlnet-mid model

The xlnet model's pre-training task differs from Bert's, employing a universal auto-regressive pre-training method. It's performance competes with Bert in many tasks, and its key innovations include bidirectional self-attention mechanisms (*Talmor et al., 2020*), the Transformer XL structure, and a larger quantity of high-quality training data. We aimed to study this model's performance in constructing taxonomy trees. Experiments reveal that xlnet's performance is second only to the highest F1 score, and we attribute these results primarily to the increase in training data.

### Roberta-wwm-ext-large model

The roberta-wwm-ext-large model is widely used and excels in various tasks within the Chinese domain. It is based on Bert but undergoes more extensive and fine-tuned training with optimized hyperparameters. Precision values of the Roberta model significantly improve when gloss is added, validating the effectiveness of gloss in enhancing accuracy. This results in a more stringent classification, achieved by reducing the number of False Positives.

## Hyperparameter analysis

### Learning rate

Learning rate is one of the most critical parameters in deep learning training. The warmup parameter also influences training effectiveness by affecting the learning rate function. Figure 4 shows the change in F1 score each epoch under different learning rates (5e-6, 1e-5, 1.5e-5, 2e-5, 5e-5) when using the Roberta model and employing a linear warmup with a warmup ratio 0.05.

We observe that the model is highly sensitive to the learning rate, and the highest point for the model is reached after the initial few epochs. Extended epochs result in F1 values oscillating at lower points, highlighting the importance of parameter selection and data quality. In the context of constructing taxonomy trees, high-quality data is crucial, and performing multiple epochs on the same batch of data offers minimal benefits. This aligns with the development direction of large language models, as noted in the work of *Touvron et al. (2023)*, which mentions training on a *corpus* of 1.4T tokens and conducting only two epochs of training on a small subset of high-quality data, and one epoch on the vast majority of data.

### Epsilon

Figure 5 illustrates the F1 values for different epsilons in each epoch. Epsilon is primarily designed for numerical stability to prevent anomalies such as divide-by-zero errors during computations. The changes in epsilon have a minor impact on the highest F1 score, suggesting that finding a local optimum in the parameter domain is a demanding condition.

## Generative annotation analysis

Figure 6 shows the changes in F1 scores with increasing epochs for ChatGPT and ChatGLM-6B for gloss extraction on the Roberta model. Table 7 presents the relative

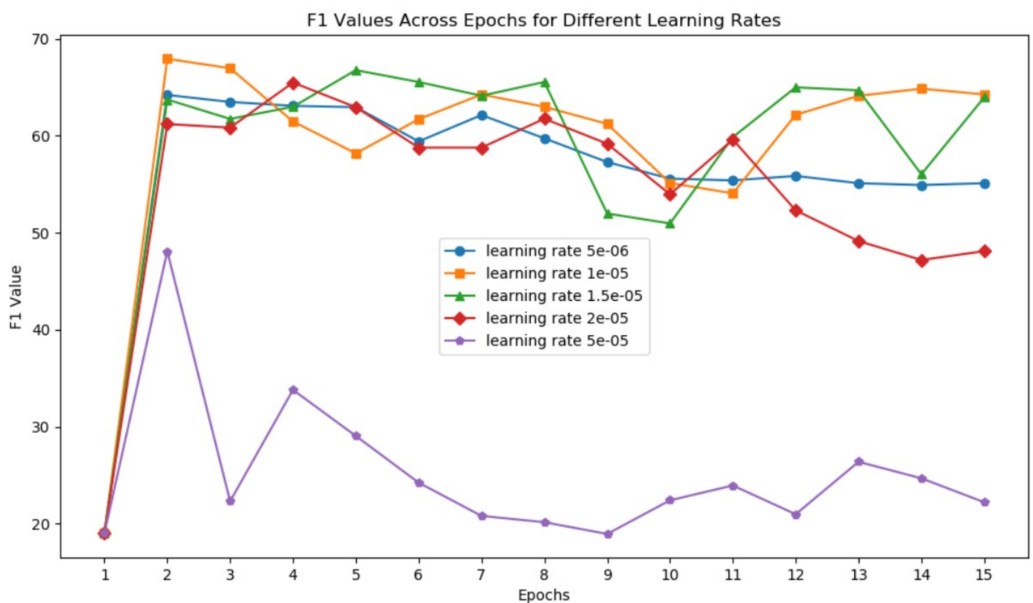

**Figure 4  F1 values across epochs for different learning rates.**

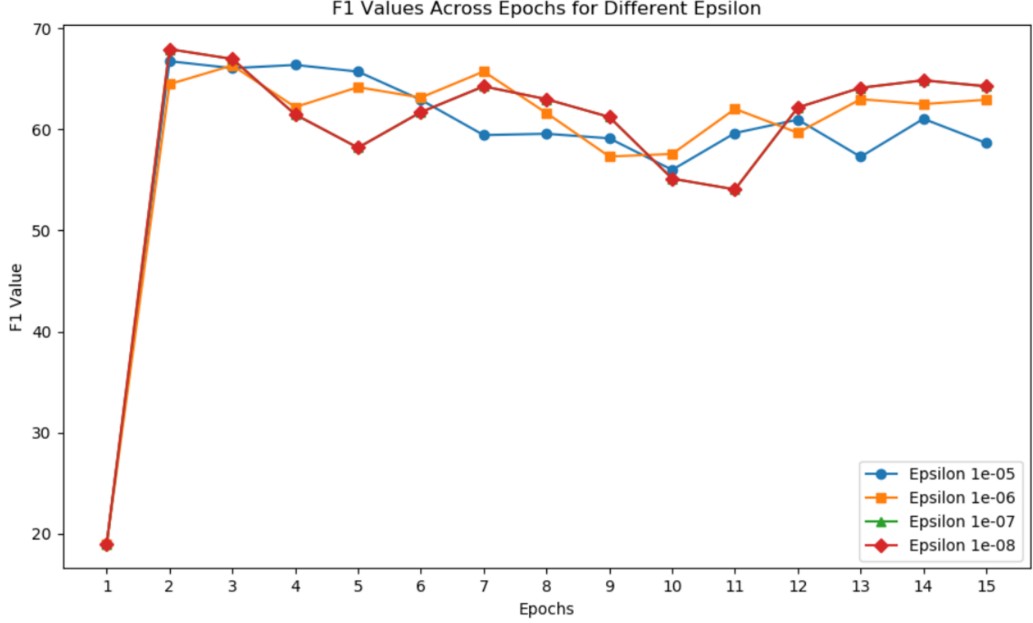

**Figure 5  F1 values across epochs for different epsilon.**

improvement in performance when using ChatGPT-generated gloss for different base models. Based on the experimental results, we draw three conclusions.

First, larger models can better leverage external knowledge. According to Table 7, adding gloss leads to a certain degree of F1 score improvement, with a 3.2% increase for Bert-base with 110M parameters and an 8.3% increase for Roberta-large with 340 M parameters. Therefore, the larger the model, the higher the relative improvement. We
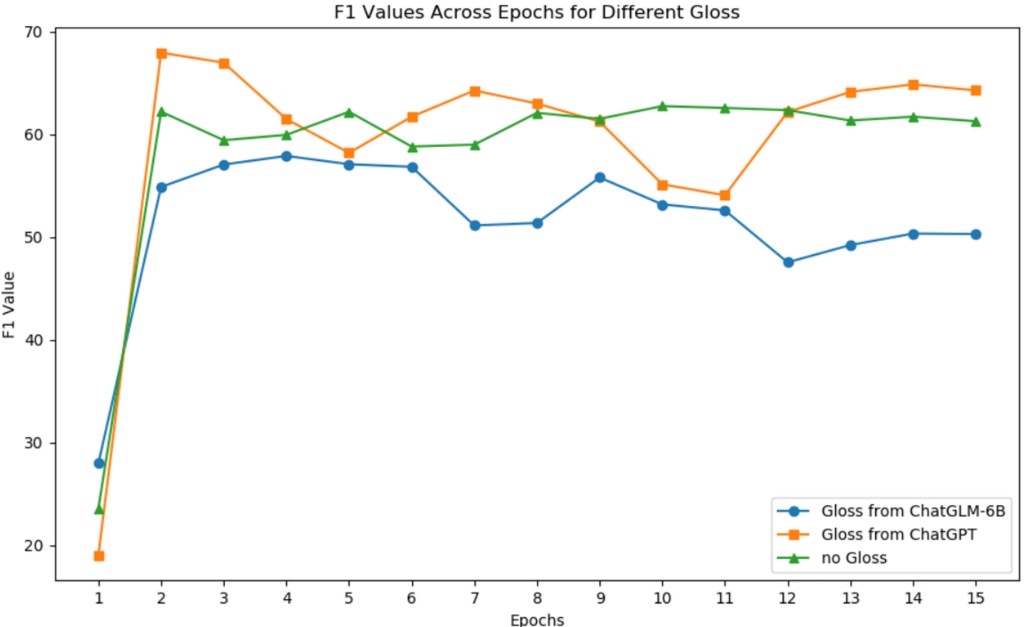

**Figure 6** F1 values across epochs for different gloss.

**Table 7 Relative improvement of different glosses.**

| Model | P | R | F1 | Relative improvement |
|---|---|---|---|---|
| Hfl-chinese-roberta-wwm-ext-large (no glosses) | 64.5 | 61.0 | 62.7 | |
| Hfl-chinese-roberta-wwm-ext-large (ChatGPT glosses) | 73.5 | 63.1 | 67.9 | 8.3% |
| Bert-base-chinese (no glosses) | 63.8 | 55.1 | 59.1 | |
| Bert-base-chinese (ChatGPT glosses) | 62.5 | 59.6 | 61.0 | 3.2% |

believe that gloss does not linearly amplify F1 or directly add to it. Its impact on F1 depends largely on the model's architecture.

Second, the quality of added gloss significantly affects F1 for hyponymy-hypernymy relationship recognition. Figure 7 illustrates the origin word "elephant" has an impact to all the token in the sentence. According to Fig. 6, ChatGPT's gloss leads to an 8.3% relative improvement in Roberta's F1 score, while ChatGLM results in a 2.4% decrease. This demonstrates that low-quality gloss can have a negative impact when inferencing. Based on *Chen, Lin & Klein (2021)* and *Peters et al. (2019)*, gloss collected from the web improves the model's performance by 3.2%. ChatGPT provides a solid baseline for gloss quality, but there is still space for improvement in gloss quality as generative large language models evolve.

Third, generative models can indirectly serve various NLP tasks. Extracting vocabulary gloss using ChatGPT is straightforward, with the training dataset consisting of only 2207 words, but it's unacceptable while processing 33 k relationships' classification. Moreover, ChatGPT's speed in defining words far exceeds that of classifying relationships. Therefore, directly extracting extensive knowledge for relationship classes using ChatGPT is not

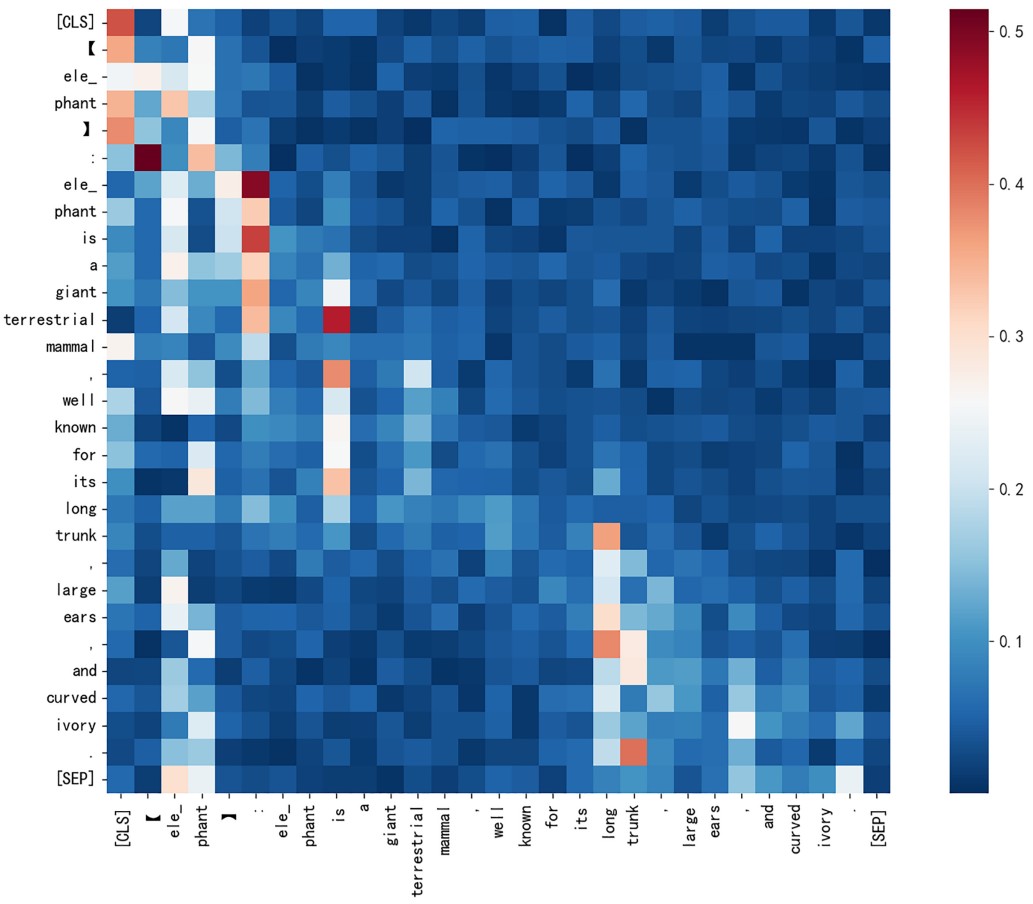

**Figure 7 The attention score heatmap from CHRRM's classification model for a sentence about elephant.**

feasible at this stage. A feasible approach is to employ ChatGPT to extract high-quality gloss as knowledge and use smaller models for relationship classification, as implemented in our work. This approach indirectly enhances performance through ChatGPT. We believe that in areas where ChatGPT may not excel, it can also provide indirect assistance by breaking down tasks into more atomic components.

## CONCLUSIONS

In this article, we addressed the task of constructing taxonomy trees in the Chinese domain. We improved the implementation of constructing taxonomy trees based on pre-trained models by (1) optimizing the selection and configuration of pre-trained models and (2) incorporating annotations obtained from generative language models. As a result, we increased the F1 score for this task from 58.7 to 67.9. We analyzed the impact of various parameters on the task of constructing taxonomy trees and discussed the feasibility of applying generative language models to this task. The following conclusions were drawn:

1) The Roberta-wwm-ext-large model consistently achieved excellent results in this type of task and could effectively extract knowledge encoded within the model through various tasks.

2) Vocabulary gloss generated by generative language models can help pre-trained models improve the accuracy of hypernym recognition. However, this improvement is limited and requires certain generation quality requirements and computational resources.

3) Generative large language models can serve various NLP tasks either directly or indirectly; it is feasible to improve the NLU task's performance through the generative content.

In conclusion, constructing taxonomy trees is a challenging yet valuable task. The emergence of generative language models has revolutionized various tasks in the NLP field. Different languages are carriers of distinct societies and cultures, and the knowledge structures encoded within languages also vary. Therefore, more work is needed in the Chinese domain to achieve superior results based on advanced technology.

### Funding
This work was supported by the Technology Development Project of China Railway Eryuan Engineering Group Co., Ltd. under Grant No.KYY2019003(19-22). The funders had no role in study design, data collection and analysis, decision to publish, or preparation of the manuscript.

### Grant Disclosures
The following grant information was disclosed by the authors:
Technology Development Project of China Railway Eryuan Engineering Group Co., Ltd: KYY2019003(19-22).

### Competing Interests
The authors declare that they have no competing interests.

### Author Contributions
- Jianyu Guo conceived and designed the experiments, performed the experiments, analyzed the data, performed the computation work, authored or reviewed drafts of the article, and approved the final draft.
- Jingnan Chen analyzed the data, prepared figures and/or tables, authored or reviewed drafts of the article, and approved the final draft.
- Li Ren performed the experiments, prepared figures and/or tables, and approved the final draft.
- Huanlai Zhou conceived and designed the experiments, performed the experiments, prepared figures and/or tables, and approved the final draft.
- Wenbo Xu analyzed the data, authored or reviewed drafts of the article, and approved the final draft.

- Haitao Jia conceived and designed the experiments, authored or reviewed drafts of the article, and approved the final draft.

## Data Availability

The code and raw data are available in the Supplemental Files.

The CTP code is also available at GitHub: https://github.com/cchen23/ctp.

## Supplemental Information

Supplemental information for this article can be found online at http://dx.doi.org/10.7717/peerj-cs.2358#supplemental-information.

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
