# Peer review of "Constructing Chinese taxonomy trees from understanding and generative pretrained language models"

_PeerJ Computer Science, doi:10.7717/peerj-cs.2358_

## Round 0.1 · original submission · Minor Revisions

Dear authors,

Thank you for submitting your manuscript to PeerJ Computer Science.

We have completed the evaluation of your manuscript. The reviewers recommend reconsideration of your manuscript following major revision. I invite you to resubmit your manuscript after addressing the comments made by the reviewers. In particular:

1. Motivate the selection of the Chu-Liu-Edmonds algorithm. Proved a brief description of this algorithm.
2. The changeable hyperparameters are listed in Table 2 and constant hyperparameters are listed in Table 3. Please, describe how hyperparameters were selected.

Please, also address the following issues:

3. The title is not grammatically correct. Update it as follows:
[add “a”] CHRRM: Constructing a Chinese taxonomy tree from understanding and generative pretrained language models
or
[replace “tree” by “trees”] CHRRM: Constructing Chinese taxonomy trees from understanding and generative pretrained language models

4. For each citation of the form Authors (year) make sure there is a space between “Authors” and “(year)”. For instance “Changlong et al.(2020)” should be “Changlong et al. (2020)”

5. Page 9: mechanisms(Alon et al.,2020) -> mechanisms (Alon et al.,2020)

I hope you can complete the recommended changes in a revision of your article.

Reviewer 1 ·

Basic reporting

This article presents a new two-stage approach for hypernym taxonomic trees construction task in the Chinese language domain. The first stage, hypernymy recognition stage utilizes the popular pre-trained models and generative large language models like BERT and ChatGPT to obtain more precise word embedding, then conduct classification based on these embeddings to obtain logits that serve as weight. The second stage, reconciliation stage utilizes Chu-Liu-Edmonds algorithm to find the maximum spanning tree of the word graph from the first stage.

The paper employs the Chu-Liu-Edmonds algorithm for forming a maximum spanning tree, yet there is no explanation provided about why this particular algorithm was selected over other possible algorithms. A detailed introduction to the Chu-Liu-Edmonds algorithm and a rationale for its selection would be beneficial.

Experimental design

The article conducts experiments on WORDNET datasets to demonstrate its performance. But the discussion on hyperparameter configuration is somewhat lacking in depth, especially in comparison to existing research. It would be beneficial to provide a more detailed description of how these hyperparameters were chosen and optimized.

Validity of the findings

This article proposes a method for constructing hypernym taxonomic trees and demonstrates the outstanding performance of GPT and BERT models in such tasks.

Additional comments

None.

·

Basic reporting

This paper proposes a new method, CHRRM, for constructing a word taxonomy in the Chinese domain. This method utilizes pretrained models to predict hypernym relationships between word pairs and generates a taxonomy through a maximum spanning tree algorithm. It also explores the use of LLM to annotate words, aiming to expand word's definitions.

points to be improved:
Some images are not in vector format, resulting in low resolution.

Experimental design

The paper’s experimental part need more context and comparisions. It’s limited to the WordNet dataset.

points to be improved:
1. The paper lacks an introduction and comparison with other related work, particularly in the experimental section, including the datasets they selected, experimental methods, and model performance.

2. Has the consideration been given to metrics other than the F1 score, such as graph similarity or graph edit distance?

Validity of the findings

The gloss generated by LLM can be used for BERT to predict hypernym relationships. Optimizing hyperparameters and base models, using gloss can obtain better taxonomy.

Additional comments

No comment.

---

## Round 0.2 · accepted · Accept

Thank you for your contribution to PeerJ Computer Science and for addressing all the reviewers' suggestions. We are satisfied with the revised version of your manuscript and it is now ready to be accepted. Congratulations!

Reviewer 1 ·

Basic reporting

The authors have well addressed all the comments!

Experimental design

Well done.

Validity of the findings

Well done.

Additional comments

No more comments!